# Chaperone-Mediated Autophagy Markers LAMP2A and HSPA8 in Advanced Non-Small Cell Lung Cancer after Neoadjuvant Therapy

**DOI:** 10.3390/cells10102731

**Published:** 2021-10-13

**Authors:** Tereza Losmanova, Philipp Zens, Amina Scherz, Ralph A. Schmid, Mario P. Tschan, Sabina Berezowska

**Affiliations:** 1Institute of Pathology, University of Bern, 3008 Bern, Switzerland; tereza.losmanova@pathology.unibe.ch (T.L.); philipp.zens@pathology.unibe.ch (P.Z.); mario.tschan@pathology.unibe.ch (M.P.T.); 2Graduate School for Health Science, University of Bern, 3012 Bern, Switzerland; 3Department of Medical Oncology, Inselspital University Hospital Bern, 3010 Bern, Switzerland; amina.scherz@insel.ch; 4Department of General Thoracic Surgery, Inselspital University Hospital Bern, 3010 Bern, Switzerland; Ralph.Schmid@insel.ch; 5Department of Laboratory Medicine and Pathology, Institute of Pathology, Lausanne University Hospital and University of Lausanne, 1011 Lausanne, Switzerland

**Keywords:** lung cancer, non-small cell lung cancer, neoadjuvant therapy, chaperone-mediated autophagy (CMA), LAMP2A, HSC70, HSPA8

## Abstract

In recent years autophagy has attracted the attention of researchers from many medical fields, including cancer research, and certain anti-macroautophagy drugs in combination with cytotoxic or targeted therapies have entered clinical trials. In the present study, we focused on a less explored subtype of autophagy, i.e., chaperone-mediated autophagy (CMA), with the key proteins LAMP2A and HSPA8 (HSC70), and their immunohistochemical evaluation with previously extensively validated antibodies. We were interested in whether the marker expression is influenced by the antecedent therapy, and its correlation with survival on a cohort of patients with non-small cell lung cancer (NSCLC) after neoadjuvant therapy and matched primary resected tumors. In concordance with our previous study, we did not find any intratumoral heterogeneity, nor correlation between the two parameters, nor correlation between the markers and any included pathological parameters. Surprisingly, the expression of both markers was also independent to tumor response or administered neoadjuvant treatment. In the survival analysis, the results were only significant for LAMP2A, where higher levels were associated with longer 5-year overall survival and disease-free survival for the mixed group of adenocarcinomas and squamous cell carcinomas (*p* < 0.0001 and *p* = 0.0019 respectively) as well as the squamous cell carcinoma subgroup (*p* = 0.0001 and *p* = 0.0001 respectively). LAMP2A was also an independent prognostic marker in univariate and multivariate analysis.

## 1. Introduction

Despite changes in smoking habits, non-small cell lung cancer (NSCLC) continues to be the leading cause of cancer-related deaths worldwide in both men and women [1]. In past years therapy has evolved, especially for cancers with known oncogenic driver mutations such as *EGFR*, *ALK*, *ROS1* and *BRAF,* or newly discovered NSCLC drivers such as *MET*, *RET*, *NTRK1/2/3* and *ERBB2 (HER2)*. It is also important to note the new emerging therapy method for tumors harboring *KRAS* p.G12C, the most common oncogenic driver mutation in NSCLC [2]. Yet, taking into account the whole group of NSCLC including squamous cell carcinoma, there is still a large proportion of cancers lacking targetable mutations [3,4]. Patients also profit from immunotherapy and immune checkpoint inhibitors, which have recently been approved for unresectable stage III cancer therapy [5,6]. Even with the unquestionable progress in treatments, the overall survival remains poor [7] and novel therapies, especially for patients with advanced cancers, are in high demand.

In stage I and II NSCLC, surgery still offers the best chance for a cure [8]. However, patients remain at high risk for recurrence and death. Because randomized trials have shown that additional neoadjuvant or adjuvant chemotherapy can lead to better outcomes compared with resection alone, it is generally accepted that patients with NSCLC in stages II-III should not be treated with surgery alone [9]. Postoperative adjuvant cisplatin-based chemotherapy increases 5-year survival rates by approximately 5% and is recommended in patients with completely resected stage II to IIIA disease and selected patients with stage IB disease, weighing the benefits and risks [10]. Meta-analyses in resectable NSCLC suggest that overall survival is longer with preoperative chemotherapy than with surgery alone [11,12]. Perioperative therapeutic approaches are a hotly debated topic, with immunotherapy-based combinations and targeted treatments in *EGFR*-mutated NSCLC dominating the current trial landscape.

Autophagy plays an important role in human physiology, specifically in maintaining cellular homeostasis and adaptation to metabolic demands. Autophagy, if dysregulated, is therefore associated with the development of many human diseases such as neurodegenerative disorders, inflammatory diseases, autoimmune diseases and cancer [13]. The role of autophagy in cancer, until now best described for macroautophagy, is complex due to its divergent impact on tumorigenesis and its dependence on the stage of the disease. In the initial process of tumor development, autophagy acts tumor suppressive by inhibiting necrosis-mediated inflammation, maintaining genome integrity and inducing autophagy-mediated cell death and senescence [14]. On the other hand, autophagy promotes cancer growth and sustains tumor cell metabolism in already established tumor cells with high metabolic turnover. This is a characteristic that might be used as a future therapeutic target, as it is observed in different tumors including non-small cell lung cancer (NSCLC), and demonstrated in preclinical studies [15,16].

Important to know for future autophagy-based therapies is the fact that there are three main types of autophagy: macroautophagy, microautophagy and chaperone-mediated autophagy (CMA). Macroautophagy has received substantial attention over the past decades and is characterized by the generation of double-membrane vesicles (autophagosomes) that engulf the substrate and fuse with the lysosome for substrate degradation inside the autolysosome. Macroautophagy can be either nonselective or selective, e.g., targeting damaged organelles or invasive microbes [17]. The main characteristic of microautophagy is the direct engulfment of cytoplasmic particles by lysosomal membranes with autophagy tubes and vesicle formation and its degradation inside of the lysosome. Microautophagy is predominantly nonselective in mammalian cells [18].

CMA is the focus of this study. It differs from micro- and macroautophagy in that it is a highly selective non-vesicle-based degradation pathway, mostly observed in mammalian cells. The targeted substrate (cargo) is transported to the cytosol of the lysosomes in a multistep process (Figure 1). First, the KFERQ (Lys-Phe-Glu-Arg-Gln)-like amino acid protein motif is recognized by cytosolic chaperones including the heat shock cognate 71kDa protein (HSC70, better known as heat shock protein family A member 8 (HSPA8)). Then, the chaperone–protein complex, in cooperation with the co-chaperones, is delivered to the lysosome, where it interacts with the receptor lysosome-associated membrane protein type 2A (LAMP2A), which transfers the oligomers into the lysosome for further degradation by lysosomal proteases [19]. As of now, two major signaling pathways regulating CMA activity have been described: mTORC2 and p38 MAPK pathways [20]. It is important to point out that all three autophagy pathways collaborate closely and that their functions are at least partly complementary to each other [21,22].

Research on the role of CMA in tumor development has focused on LAMP2A, a key protein in CMA. In mouse models, *LAMP2a* knockdown in the NSCLC xenografts sensitized the tumor cells to chemotherapy; on human tissue, higher LAMP2A immunohistochemical expression was observed in non-responder patients [23]. These facts could be potentially useful in chemotherapy management, e.g., as biomarkers to predict the patient’s therapy response, or as a therapy option in chemoresistant tumors. Expression data of CMA-associated molecules in tumors are still limited, and further studies to assess CMA levels in human tumors will elucidate these hypotheses.

The aim of the present study was to evaluate the expression of the CMA-associated proteins LAMP2A and its co-player HSPA8 in locally advanced resected human NSCLC and the dependence of the expression from antecedent neoadjuvant therapy.

## 2. Materials and Methods

### 2.1. Patient Cohort

The patient collective of this retrospective single-center study consisted of a study cohort and a control cohort. The study cohort consisted of 130 consecutive NSCLC, resected after neoadjuvant treatment and diagnosed at the Institute of Pathology of the University of Bern between 2000 and 2016, including 64 adenocarcinomas (LUAD) and 58 squamous cell carcinomas (LUSC), 3 carcinomas with adenosquamous (LUASC) morphology, 2 neuroendocrine carcinomas and 4 carcinomas not otherwise specified. The control cohort consisted of biologically matched primary resected carcinomas, i.e., 60 LUAD and 55 LUSC with mediastinal lymph node metastases, which would have led to neoadjuvant treatment if the metastases had been known before resection. On a side note, one patient had in addition to a large LUSC a small LUAD, irrelevant for survival statistics. In the subcohort of untreated LUAD, the solid growth pattern was the most predominant pattern (48%), followed by micropapillary (26%), acinar (22%) and papillary (4%) morphology.

For the purposes of this study, all tumors were restaged according to the current UICC TNM classification 2017 (8th edition) [24]. Tumor typing was retrospectively validated according to current guidelines [25]. Tumor regression was graded into four categories (<1%, 1–10%, 11–49%, ≥50% of residual viable tumor) as previously described [26]. Therapy-induced changes were defined as tumor necrosis, inflammation including xanthogranulomatous reaction and fibrosis [27]. Finally, the database was completed with clinical and follow-up information by consulting the clinical files and by contacting the cantonal cancer registry and general practitioners. Three patients could not be included in the final cohort due to missing tissue and two patients were excluded due to neuroendocrine histology (large cell neuroendocrine carcinomas). For a further 25 patients, immunohistochemical evaluation was not possible. This resulted in a total of 215 patients (study cohort: *n* = 101, control cohort: *n* = 114) for comparison of autophagy marker expression.

From the study cohort, 41 (19%) patients received at least 1 cycle of platinum-based chemotherapy and 50 (23%) patients were treated according to the optimal regimen of Inselspital, which consists of at least 3 cycles of platinum-based chemotherapy and taxane. In addition, 10 (5%) patients received preoperative therapy, but we could not retrospectively validate the neoadjuvant intent. Additional radiotherapy was administered in 24 (11%) patients.

For survival analyses, we excluded patients with systemic treatment prior to resection but without neoadjuvant intention (*n* = 10), stage IV disease (*n* = 14), extra-anatomical resection (*n* = 2) or perioperative death defined as occurring within 30 days after resection (*n* = 11). Due to the multimorbidity of the cohort, we considered only the 5-year survival rate. The median overall survival (OS), which refers to the duration of survival after the start of the treatment (i.e., start of neoadjuvant regimen or resection), was 35 months (95% CI 29–NA), with 86 events reported. The median disease-free survival (DFS), which refers to the time from the start of treatment to loco-regional relapse, distant metastases or death, was 18 months (95% CI 15–25) with 116 reported events.

The study was performed according to the REMARK guidelines, and it was approved by the Cantonal Ethics Commission of the Canton of Bern (KEK 2017-00830), which waived the requirement for written informed consent.

### 2.2. Next Generation Tissue Microarrays (TMA)

The most suitable and preserved formalin fixed paraffin embedded (FFPE) block with sufficient tumor tissue was selected for each tumor. The respective hematoxylin and eosin (H & E) slide was scanned and digitally annotated by a pathologist specialized in pulmonary pathology (SB). For each patient, at least four punches (diameter = 0.6 mm) were randomly chosen from different tumor regions, including the tumor center and the infiltration zone. The cores of the chosen regions were automatically transferred from the “donor” blocks into the “recipient” TMA block, using the TMA Grandmaster (Budapest, Hungary) [28]. The cores from each tumor were placed on two different TMA blocks to prevent technical bias when performing and evaluating immunohistochemical staining.

### 2.3. Immunohistochemical Staining and Scoring

Proper validation of the specificity of antibodies used in biomarker research is very important [29]. We have previously comprehensively validated both markers on different cell lines using knock down and overexpression experiments, Western blotting and immunohistochemical staining of FFPE cell pellets [30]. Immunohistochemical staining for LAMP2A and HSPA8 was performed on 4 μm sections of TMA blocks using the automated immunostainer Leica Bond RX (Leica Biosystems, Heerbrugg, Switzerland). The following staining conditions were applied: LAMP2A (rabbit monoclonal, ab125068; Abcam, Cambridge, UK): 1:500, tris buffer, 95 °C, and 30 min; and HSPA8 (mouse monoclonal, MA3-014; Thermo Fisher Scientific, Waltham, MA, USA): 1:10,000, citrate buffer, 100 °C and 30 min. For visualization, the Bond Polymer Refine Detection kit (Leica Biosystems, Muttenz, Switzerland, DS9800) was used following the instructions of the manufacturer. Each tumor core was separately evaluated at 10× objective magnification by a pathologist experienced in the evaluation of CMA markers (TL) [30]. Each core was assigned a numerical value depending on the intensity of the cytoplasmic staining (0—negative, 1—weak, 2—medium, 3—strong) and the percentage of stained tumor cells (0 ≤ 5%, 1 = 6–25%, 2 = 26–50%, 3 = 51–75%, 4 = 76–100%) (Figure 2 and Figure 3). For individual cores, the immunoreactivity score (IRS) was calculated by multiplying the numerical values of the percentage times the intensity. The mean IRS over all assessed cores for a tumor case was considered as case specific IRS. This allowed a semiquantitative estimation of the marker expression level [31]. Nuclear staining of LAMP2A and HSPA8 was not considered in statistical analyses as only 2 and 3 cores, respectively, showed nuclear positivity.

Due to technical staining errors or insufficient amount of tissue present on the slide, evaluation of stainings was possible in 1397 cores for LAMP2A and in 1382 cores for HSPA8. The corresponding IRS could therefore be calculated for 216 cases.

### 2.4. Statistical Methods

All statistical analyses were performed using R software (version 4.0.5, https://cran.r-project.org, accessed on 1 April 2021) with suitable packages. To assess the heterogeneity of marker expression, we used the Friedman and Wilcoxon signed-rank test. For the assessment of intercore heterogeneity, only cases with at least 4 assessed cores were included. For the assessment of association between clinicopathological parameters and the expression of autophagy markers, the Wilcoxon rank-sum test, Mantel–Haenszel test and logistic regression were used. In order to dichotomize autophagy marker expression in low and high expression, we used the maximally selected rank statistics applying log-rank scores as test statistic and approximating the *p*-value according to Hothorn and Lausen in survival cohort [32]. Kaplan–Meier plots were used for the visualization of survival data including the corresponding *p*-value according to the log-rank test. Cox regression was used for univariate and multivariate analysis. A two-sided level of significance at *p* = 0.05 was considered statistically significant.

## 3. Results

### 3.1. No Significant Intratumoral or Region-Specific Heterogeneity of LAMP2A and HSPA8

All cases with at least four evaluable cores per tumor were used for assessment of heterogeneity of marker expression throughout the tumor, resulting in 197 cases for LAMP2A and 196 cases for HSPA8. For cases with more than four evaluated cores, four cores were randomly picked (considering both tumor center and infiltration zone). There was no overall heterogeneity (LAMP2A *p* = 0.6615, HSPA8 *p* = 0.4932). In order to assess the region-specific heterogeneity, the mean IRS of the tumor’s center and the infiltration zone available in total for 97 (LAMP2A) and 95 cases (HSPA8) were compared. LAMP2A expression was significantly higher in cores of the infiltration zone (*p* = 0.0056). However, we observed no significant difference for HSPA8 (*p* = 0.4972). These results must be interpreted carefully, as the number of cores evaluated for the corresponding regions was very variable, with a higher number of samples originating from the tumor center (median: 6, range 1–12; infiltration zone median: 2, range 1–4, see Appendix A).

### 3.2. No Correlation between LAMP2A and HSPA8 Expression

The expression of the two markers LAMP2A and HSPA8 did not correlate, neither on the core level (*p* = 0.0863) nor on the case level (*p* = 0.7888) for the whole cohort. Neither was there a correlation of marker expression in the subgroups of NSCLC resected after neoadjuvant therapy (*p* = 0.976), nor primary resected tumors (*p* = 0.842), nor in the histological subgroups (LUAD *p* = 0.340, LUSC *p* = 0.648).

### 3.3. Association of LAMP2A and HSPA8 Expression Levels with Pathological Parameters and Preoperative Chemotherapy

There was no correlation of LAMP2A or HSPA8 expression with the age of the patient at time of surgery (LAMP2A *p* = 0.948; HSPA8 *p* = 0.189) or patient’s gender (LAMP2A *p* = 0.273; HSPA8 *p* = 0.214).

To test for a possible selectivity of marker expression for different histological NSCLC tumor types or association to preceding chemotherapy, we excluded three adenosquamous carcinomas (LUASC) due to low sample size. Neither the IRS for LAMP2A expression in the whole cohort significantly differed between histologies (*p* = 0.384, 100 LUSC and 112 LUAD) nor the LAMP2A expression after correcting for systemic treatment before resection (*p* = 0.446, neoadjuvant 42 LUSC and 46 LUAD; *p* = 0.146, primary resected 54 LUSC and 60 LUAD). Similar results were observed for HSPA8 expression, showing no effect of the underlying histological type on marker expression (*p* = 0.284 whole cohort, *p* = 0.775 neoadjuvant, *p* = 0.531 primary resected).

We performed the same analyses based on the differences in treatment before specimen recovery. We analyzed whether no treatment at all could influence the expression of LAMP2A (*p* = 0.223) or HSPA8 (*p* = 0.895). We also excluded cases in which patients received preoperative treatment without neoadjuvant intention (*n* = 10). In all scenarios, neither LAMP2A (*p* = 0.19) nor HSPA8 expression (*p* = 0.988) were influenced by preoperative exposition to cytotoxic agents.

Furthermore, there was no association between LAMP2A (*p* = 0.609) or HSPA8 (*p* = 0.74) and the TNM tumor stage merged into four categories (stage I, stage II, stage III, stage IV), which was only examined in the neoadjuvant cohort. We also investigated the influence of the tumor bed size on the expression of LAMP2A and HSPA8, which resulted in no significant effect.

An important prognostic marker in NSCLC after neoadjuvant treatment is the proportion of residual tumor cells in the original tumor bed [33]. It is a marker of tumor response to the neoadjuvant treatment and is also used as an end point in clinical studies. Neither LAMP2A (*p* = 0.68) nor HSPA8 (*p* = 0.997) expressions were significantly associated with the regression grade. Furthermore, tumors showing major pathological response (LUSC ≤ 10% and LUAD ≤ 65% residual tumor) [26] showed similar marker expression.

Treatment-naïve LUAD (primary resected) can be stratified according to their predominant growth patterns (lepidic, acinar, papillary, micropapillary, solid) which are associated with the prognosis [34]. Purely lepidic tumors ≤ 3 cm diameter represent in situ carcinoma; acinar and papillary tumors are considered low grade; and micropapillary and solid are considered high-grade tumors. Due to only two patients with a predominant papillary growth pattern, papillary and acinar carcinomas were merged in only one class. No carcinomas with predominant lepidic growth pattern were present in the cohort. Overall, the LAMP2A expression was lower in solid LUAD compared to the other growth patterns (*p* = 0.028). In the post-hoc analysis, only the difference between papillary/acinar and solid cancers remained statistically significant (*p* = 0.034). There was no difference in HSPA8 expression (*p* = 0.181).

Molecular data from routine analyses were available for 5 LUSC and 42 LUAD cases and 1 LUASC case. Due to the long period of inclusion, different methods were used (Next Generation Sequencing, Sanger Sequencing, and fluorescence in situ hybridization). There was no association between the known mutations (including *EGFR*, *ALK*, *ROS*, *KRAS*, *TP53* or *HER2*) and any of the two markers. Table 1 shows the basic clinicopathological characteristics of the study cohort (resected after neoadjuvant treatment) and the control cohort (primary resected with mediastinal lymph node metastases) in relation to LAMP2A expression.

### 3.4. Correlation with Survival (OS and DFS)

In a three-tier classification based on quartiles cut-offs (low = 1st quartile, intermediate = 2nd and 3rd quartiles, high = 4th quartile), a higher LAMP2A expression was associated with longer OS in the whole collective (*p* = 0.02) and in primary resected LUSC (*p* = 0.0022). Prognostic significance for OS of LAMP2A was not shown in LUAD (*p* = 0.42) nor in cases after neoadjuvant treatment, irrespective of histology (*p* = 0.83 all patients, *p* = 0.97 LUSC, *p* = 0.71 LUAD). HSPA8 was not a prognostic marker for OS in any of the studied groups.

Subsequently, maximally selected rank statistics were used to dichotomize LAMP2A and HSPA8. For HSPA8, it was not possible to determine a dichotomizing cut-off for survival. For LAMP2A, a cut-off at an IRS of 7.43 was determined defining high expressing cases by an IRS > 7.43 and low expressing cases by an IRS ≤ 7.43. The LAMP2A cut-off was prognostic for OS in the whole cohort (*p* < 0.0001) and in the subgroup of primary resected LUSC (*p* = 0.0001) (Figure 4). Lower LAMP2A expression seemed to be also associated with a shorter survival in all the other subgroups; however, it was not statistically significant.

Because HSPA8 was not prognostic for OS, we excluded it from evaluation of the prognostic impact on DFS. OS is considered a hard endpoint and is much more reliable than DFS, especially in retrospective studies, thus a potential prognostic effect of HSPA8 would be highly questionable. Furthermore, only 30 additional events were counted for the entire cohort. Higher LAMP2A expression was associated with longer DFS in the whole cohort (*p* = 0.0019) and in the subgroup of primary resected LUSC (*p* = 0.00099) (Figure 4).

Multivariate analysis including age, gender, treatment group, pT category, surgical procedure, resection status and histology confirmed higher LAMP2A to be an independent prognostic marker for OS in the entire cohort (*p* = 0.006; HR 0.52; CI 0.33–0.83) and in primary resected LUSC (*p* = 0.002; HR 0.23; CI 0.091–0.6) (Figure 5). This remained true for DFS (Appendix A).

## 4. Discussion

The role of autophagy and particularly CMA in cancer development and treatment response is complex. Therefore, a better understanding of this process is warranted before cancer biomarkers or therapies based on autophagy modulation can be considered. In most human cancers, including lung cancer, CMA is upregulated [35], implying an important role of CMA in tumor biology. Both tumor suppressive and pro-oncogenic effects of CMA have been described, depending on specific type and stage of the tumor. Under normal conditions, CMA maintains the balance of cell cycle regulators, and therefore works as tumor suppressive [36]. On the other hand, the tumor protective role is attributed to several mechanisms, among others bridging the time of cellular stress by anaerobe tumor cell metabolism, protection to hypoxic stress and involvement in tumor migration ability [37]. It is also worth noting that under stress conditions, including tumor development, tumor cell survival is not only dependent on autophagy but also on a delicate interplay between autophagy and apoptosis [38].

The available and established methods for examining autophagy include transmission electron microscopy, Western blotting, flow cytometry, immunofluorescence and immunohistochemistry (IHC) [39]. Due to the dynamic nature of autophagy in the cell (autophagic flux), functional assays are the most suitable methods for measuring autophagic activity. However, these assays cannot be performed on FFPE tissue, which is available for retrospective analysis and in routine diagnostic pathology practice. The immunohistochemical expression analysis of autophagy-associated molecules is suitable for examination on FFPE material, and corresponds to the capture of autophagy at a particular point in time. This snapshot does not necessarily represent the actual flux, even though it is suggestive for the autophagy level. The obvious advantage of IHC is the applicability to large collectives of patients. Furthermore, IHC can be processed and evaluated in any pathology laboratory and could potentially be used as a method to assess CMA biomarkers in clinical studies, or in routine cancer diagnostics.

For the purpose of our study, we used previously extensively validated antibodies of LAMP2A and HSPA8 to evaluate their expression in NSCLC, accounting for the strength of the study. Both LAMP2A and HSPA8 showed no correlation to any of the studied pathological parameters, nor any association to each other, which aligned with our previous study results [30]. The expression was also unrelated to the underlying tumor histology.

Although both markers closely cooperate in the CMA process, their role and localization in the cell is different. HSPA8 belongs to the heat shock protein family, is located in various cellular areas and is involved in CMA and general protein maintenance, apoptosis and cellular signaling [40]. On the other hand, LAMP2A is exclusively found in the lysosome and is the only isoform of LAMP2 associated with CMA, representing its rate-limiting factor [41].

Compared to our preceding study, HSPA8 did not show any prognostic value overall, nor in any of the subgroups. LAMP2A was a prognostic marker overall and in the primary resected LUSC subgroup. Interestingly, high expression was associated with better prognosis, unlike the results of our previous study on primary resected LUSC. This difference could be explained by the different patient composition with a predominance of low stage tumors (stage I and II) in our previous study [30]. To date, most published immunohistochemical studies on the expression of LAMP2A in NSCLC have shown high expression to be associated with worse survival. The percentage of stage I and II patients in the NSCLC cohorts of these studies was as follows: 100% [42,43], 70% [44], 43% [23] with 0, 3 and 0 patients in stage IV, respectively. Furthermore, the dichotomous role of autophagy in cancers with tumor suppressive and pro-survival effects needs to be taken into account. Moreover, these effects are best studied in macroautophagy, and the exact role of CMA during tumorigenesis remains unclear. As mentioned above, IHC on FFPE tissue is only a snapshot in time of the whole autophagy process, and high levels can implicate activated autophagy as well as errors in its degradation or lysosomal dysfunction, warranting further functional analyses.

In our cohort, neither LAMP2A (*p* = 0.68) nor HSPA8 (*p* = 0.997) expressions were significantly associated with the histopathological regression grade. Furthermore, neither LAMP2A nor HSPA8 expression seemed to be influenced by preoperative exposition to chemotherapy.

Numerous autophagy inhibitors have been discovered. Chloroquine (CQ) and its derivative hydroxychloroquine (HCQ) block the fusion of autophagosomes with lysosomes and thus affect mainly macroautophagy [45]. Its possible influence on chemotherapy response is already being studied in clinical trials including studies on NSCLC [46]. The advantage of adding HCQ to the standard chemotherapy regimen was detected in patients with KRAS mutated tumors [47]. For the specific inhibition of CMA, namely the interaction with HSPA8, a peptide called P140 was discovered a few years ago, successfully undergoing clinical trials for the treatment of systemic lupus erythematosus [48], which may represent a promising therapeutic option in the future. When P140 or other CMA modulators will be considered for treating cancer, patient selection by means of tissue-based biomarkers will become important. Our study aimed to add data on the character, dependence from previous chemotherapy and prognostic value of CMA marker expression in advanced NSCLC tissue to the body of evidence informing biomarker evaluation. Among the limitations of our study is its retrospective design, which might have affected the patient group assessment, although the cohort was constructed according to the REMARK criteria for tumor marker testing. Additionally, the evaluation of autophagy using IHC as a static method does not measure the autophagy flux. Nevertheless, assessment of autophagy markers using IHC remains the most suitable method for the evaluation in daily routine work in pathological diagnostics, should they become biomarkers in the future. Another limitation regarding the immunohistochemical procedure might be the different age of included FFPE blocks. However, all blocks were stored according to guidelines and we could exclude any bias in staining due to storage time of the FFPE blocks (Appendix A). Furthermore, expression of CMA markers after neoadjuvant chemotherapy should be compared with pre-therapeutic, diagnostic biopsies in dedicated future studies. We had only a very limited number of pre-chemotherapy biopsies or cytologies derived from the primary tumor available for our real-life collective, and were thus not equipped to perform a direct comparison in the present study. Even though we tried to overcome this limitation by including tissue from a biologically matched primary-resected control cohort, our results warrant extension to future direct pre/post chemotherapy comparisons.

## 5. Conclusions

In conclusion, we demonstrated the independent immunohistochemical expression of CMA markers LAMP2A and HSPA8 in LUSC and LUAD. High levels of LAMP2A were associated with longer overall survival in patients with locally-advanced NSCLC. In NSCLC resected after neoadjuvant (radio-)chemotherapy, there was no correlation of CMA marker expression with antecedent therapy nor with therapy response. With the perspective of future clinical trials targeting autophagy in addition to standard treatment, further studies on expression of CMA markers in the neoadjuvant setting are warranted.

## Figures and Tables

**Figure 1 cells-10-02731-f001:**
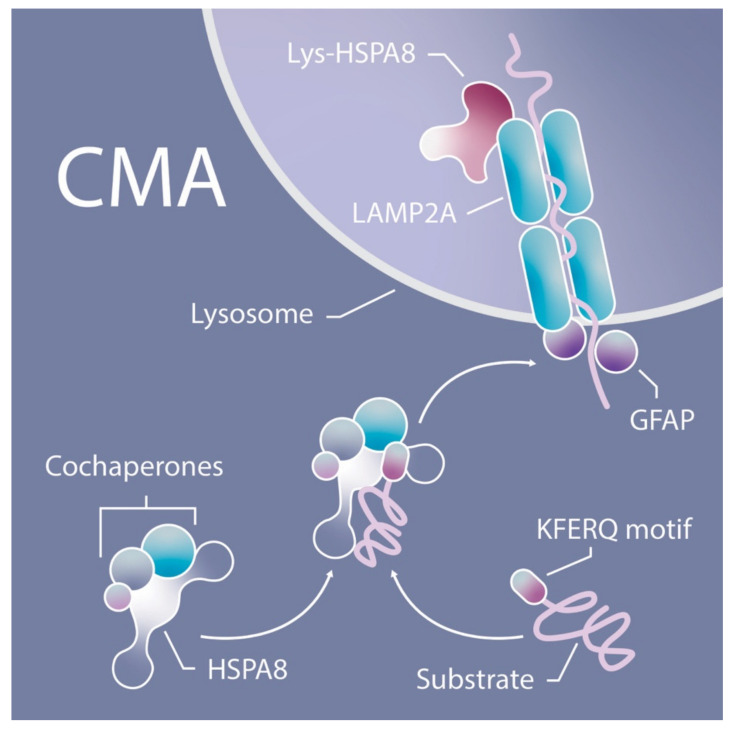
CMA. Substrate transport to the lysosome by HSPA8 and LAMP2A.

**Figure 2 cells-10-02731-f002:**
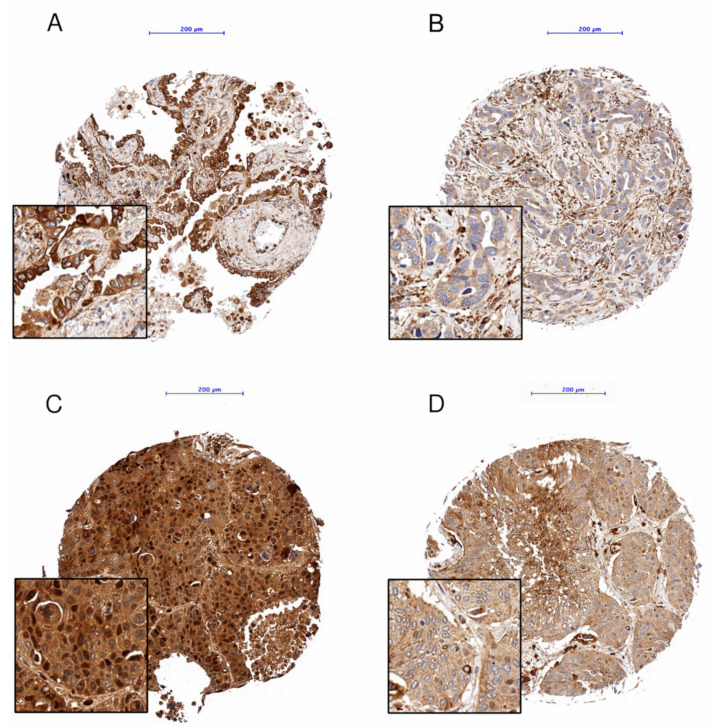
LAMP2A, examples of immunohistochemical staining, (**A**): Adenocarcinoma, IRS 3 × 4 = 12; (**B**) Adenocarcinoma, IRS 1 × 4 = 4; (**C**): Squamous cell carcinoma, IRS 3 × 4 = 12; (**D**): Squamous cell carcinoma, IRS 2 × 4 = 8; Objective magnification: 10×, Scale bar: 200 µm.

**Figure 3 cells-10-02731-f003:**
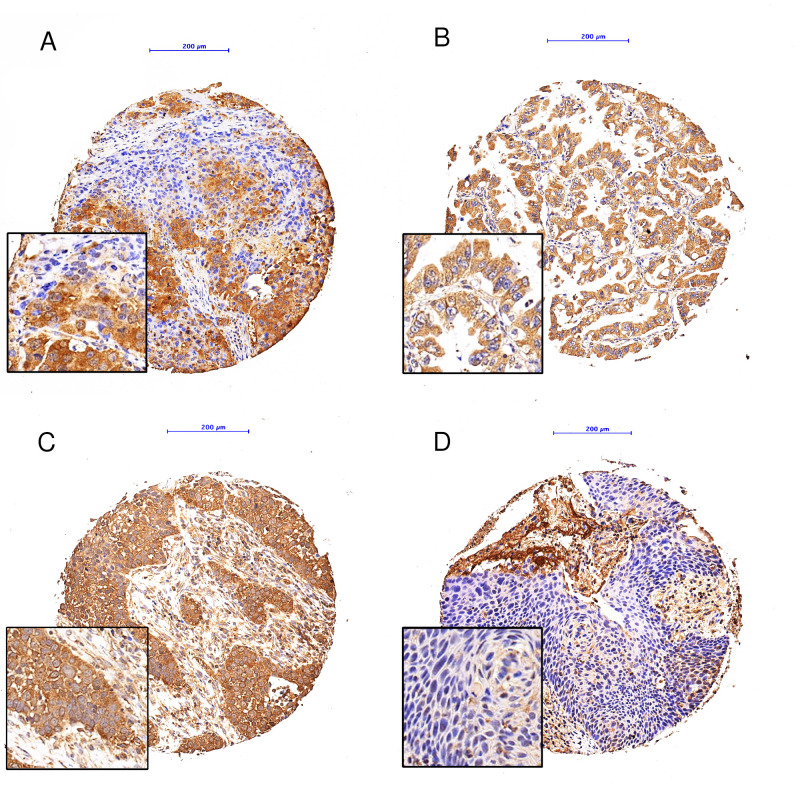
HSPA8, examples of immunohistochemical staining: (**A**): Adenocarcinoma, IRS 3 × 3 = 9; (**B**) Adenocarcinoma, IRS 2 × 4 = 8; (**C**): Squamous cell carcinoma, IRS 2 × 4 = 8; (**D**): Squamous cell carcinoma, IRS 0 × 4 = 0; Objective magnification: 10×, Scale bar: 200 µm.

**Figure 4 cells-10-02731-f004:**
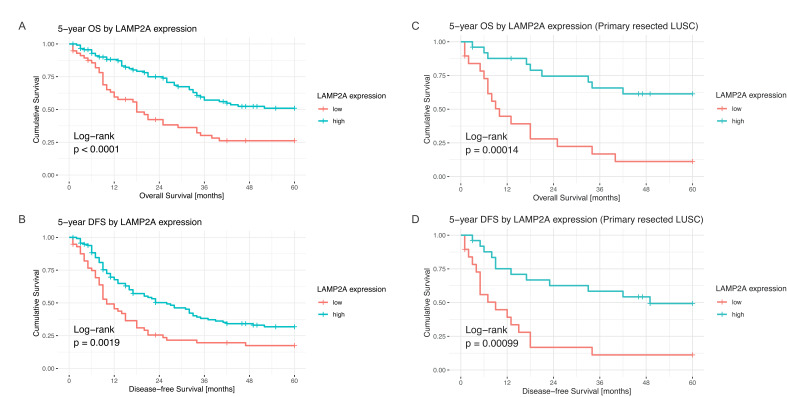
Survival curves for LAMP2A, (**A**): OS, overall; (**B**): DFS, overall; (**C**): OS, LUSC subgroup; (**D**): DFS, LUSC subgroup.

**Figure 5 cells-10-02731-f005:**
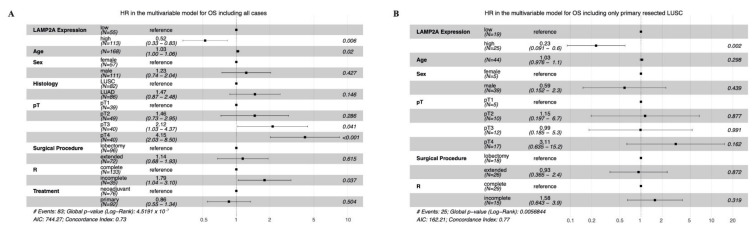
Multivariable analysis (**A**) OS, including all cases; (**B**) OS, primary resected LUSC. LUSC: Lung squamous cell carcinoma, LUAD: Lung adenocarcinoma, Extended: Pneumonectomy or bilobectomy.

**Table 1 cells-10-02731-t001:** Basic clinicopathological characteristics of the study and the control cohort in relation to LAMP2A expression. ° Mantel–Haenszel test, * logistic regression, + Wilcoxon rank-sum test.

		Study Cohort (*n* = 101)		Control Cohort (*n* = 114)		*p*-Value
		LAMP2A Low (*n* = 26)	LAMP2A High (*n* = 75)	LAMP2A Low (*n* = 42)	LAMP2A High (*n* = 72)	
Age, years (median [IQR])		64 [56–69.8]	63 [55–69.5]	63 [57–70.8]	64.5 [58.8–70]	0.945 *
Gender		*n* = 26 (%)	*n* = 75 (%)	*n* = 42 (%)	*n* = 72 (%)	0.347 °
Female		7 (26.9)	22 (29.3)	11 (26.2)	27 (37.5)	
Male		19 (73.1)	53 (70.7)	31 (73.8)	45 (62.5)	
Smoking status		*n* = 22 (%)	*n* = 63 (%)	*n* = 33 (%)	*n* = 57 (%)	0.133 °
Never/Ex-smoker		15 (68.2)	42 (66.7)	15 (45.5)	40 (70.2)	
Active smoker		7 (31.8)	21 (33.3)	18 (54.5)	17 (29.8)	
Histology		*n* = 26 (%)	*n* = 75 (%)	*n* = 42 (%)	*n* = 72 (%)	0.137 °
LUSC		11 (42.3)	35 (46.7)	24 (57.1)	30 (41.7)	
LUAD		13 (50)	39 (52)	18 (42.9)	42 (58.3)	
LUASC		2 (7.7)	1 (1.3)			
Macroscopic tumor bed, cm (median [IQR])		4.2 [3.55–5.88]	3.5 [2.5–5.25]	5.45 [3.75–7.15]	4.2 [2.85–6]	0.059 *
Resection		*n* = 26 (%)	*n* = 75 (%)	*n* = 42 (%)	*n* = 72 (%)	0.327 °
Wedge		1 (3.8)	1 (1.3)	2 (4.8)	1 (1.4)	
Lobectomy		15 (57.7)	38 (50.7)	17 (40.5)	45 (62.5)	
Bilobectomy		1 (3.8)	5 (6.7)	5 (11.9)	3 (4.2)	
Pneumonectomy		9 (34.6)	31 (41.3)	18 (42.9)	23 (31.9)	
HSPA8, IRS (median [IQR])		8 [7.38–8.67]	8 [7.46–9.33]	8 [7.36–8.67]	8 [7.29–9.14]	0.413 *
AJCC/UICC (yp)TNM stage 2017		*n* = 26 (%)	*n* = 75 (%)	*n* = 42 (%)	*n* = 72 (%)	0.805 +
Stage I		3 (11.5)	13 (17.3)			
Stage II		6 (23.1)	19 (25.3)			
Stage III		17 (65.4)	36 (48)	40 (95.2)	64 (88.9)	
Stage IV			7 (9.3)	2 (4.8)	8 (11.1)	
Regression, residual tumor		*n* = 26 (%)	*n* = 75 (%)			0.115 +
MPR	<1%	1 (3.8)	7 (9.3)			
	1–10%	2 (7.7)	10 (13.3)			
10–50%		4 (15.4)	16 (21.3)			
≥50%		19 (73.1)	42 (56)			
EGFR status		*n* = 5 (%)	*n* = 16	*n* = 9	*n* = 15	0.795 °
WT		4 (80)	13 (81.3)	8 (88.9)	11 (73.3)	
Mutated		1 (20)	3 (18.7)	1 (11.1)	4 (26.7)	
ALK status		*n* = 4	*n* = 13	*n* = 5	*n* = 11	
WT		4 (100)	13 (100)	5 (100)	11 (100)	
Mutated						
ROS1 status		*n* = 2	*n* = 13	*n* = 2	*n* = 9	0.695 °
WT		2 (100)	12 (92.3)	2 (100)	9 (100)	
Mutated			1 (7.7)			
KRAS status		*n* = 2	*n* = 13	*n* = 1	*n* = 6	0.81 °
WT		1 (50)	10 (76.9)	1 (100)	4 (66.7)	
Mutated		1 (50)	3 (23.1)		2 (33.3)	
TP53 status		*n* = 2	*n* = 12	*n* = 1	*n* = 6	0.094 °
WT			8 (66.7)		5 (83.3)	
Mutated		2 (100)	4 (33.3)	1 (100)	1 (16.7)	
HER2 status		*n* = 2	*n* = 12	*n* = 2	*n* = 6	0.198 °
WT		2 (100)	11 (91.7)		6 (100)	
Mutated			1 (8.3)	2 (100)		
R status		*n* = 25 (%)	*n* = 74 (%)	*n* = 40 (%)	*n* = 71 (%)	0.257 °
R0		19 (76)	63 (85.1)	28 (70)	55 (77.5)	
R1/R2		6 (24)	11 (14.9)	12 (30)	16 (22.5)	

## Data Availability

The data is available upon reasonable request.

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
