# Peer review of "Chaperone-Mediated Autophagy Markers LAMP2A and HSPA8 in Advanced Non-Small Cell Lung Cancer after Neoadjuvant Therapy"

_cells, 2021, doi:10.3390/cells10102731_

Round 1

Reviewer 1 Report

The authors have studied the role of two CMA markers in NSCLC.

  • First of all, I would like the authors to explain why they have considered that the study was carried out at an advanced stage.
  • Italicize all gene names.
  • Please describe the acronyms with the first use.
  • In the introduction section, first paragraph. Add reference/s of clinical practice guidelines on lung cancer.
  • Abbreviations should be checked
  • Please revise the last paragraph introduction section “The aim of the study..”
  • In my opinion, the material and methods section includes some study results. Please modify. I.e. Patients characteristics, table 1, DFS, OS, number of patients.
  • Line 137. The authors for the first time comment that patients samples will be evaluated by TMA. Considered to add this information elsewhere.
  • Line 138. Why was neuroendocrine histology excluded? Are you referring to patients diagnosed with small-cell lung cancer?
  • Add this information in Material & Methods: the study was approved by the Cantonal Ethics Commission of the Canton of Bern (KEK 420 2017-00830).
  • Line 161. What was considered sufficient tumor tissue? And the criteria to add more punches? Add the range of punches. Why were there cases with a punch representation of 16?
  • Please modify “We observed high levels of LAMP2A to be associated with favorable patients' survival in advanced stage NSCLC.”
  • Line 195. 216 cases: how many were from the control study cohort?
  • Line 274-275. Add reference.
  • Line 289. How was the cut-off determined? Figure 5 What was considered low/high?
  • Figure 5. Better quality image is needed. It is difficult to completely discern what is shown in this image.
  • The authors discuss some limitations of their proposed method. Could you comment on the age of the samples? For diagnostic purposes, it is normally recommended that samples be less than 3 years old.

Reviewer 2 Report

Major comments:

  1. Cancer cachexia, a syndrome of physical wasting and malnutrition, usually occurs at the advanced stages of cancer. Participant’s nutritional status might affect chaperone-mediated autophagy (CMA) and to interfere expression levels of CMA markers. High dietary lipid intake has been demonstrated to inhibit CMA (Proc Natl Acad Sci USA. 2012, 109(12):E705-14). Thereby, records of participant’s nutritional status and dietary intake should be included to analyze potential influence in the present study.
  2. Previous study has already used a very similar research design and strategy to examine the correlation between expression levels of two CMA markers (LAMP2A and HSPA8) and lung cancer (Oxid Med Cell Longev. 2020, 2020:8506572). Thereby, novelty of the present study is relatively less.
  3. Are there any clues suggesting a possible link between neoadjuvant therapy and CMA? Please discuss them in Introduction section.
  4. Please discuss the potential application of detecting tumoral LAMP2A level as well as how this analyzed result improves clinical outcome of lung cancer therapy.
  5. As the sentences stated by the authors, autophagy can be acted as a suppressor or a promoter for tumor progression. Please discuss whether there is a practical application for CMA stimulators or inhibitors in clinical cancer therapy.
  6. Manufacturers and catalog numbers of two critical antibodies against LAMP2A and HSPA8 proteins used in the present study are wrong. Are these primary antibodies the same like previous study (Oxid Med Cell Longev. 2020, 2020:8506572)?

Minor comments:

  1. Abbreviations should be defined at first mention.

Reviewer 3 Report

In this article, the authors aim to assess CMA markers LAMP2A and HSPA8 (using IHC staining) as predictive biomarkers to platinum-based chemotherapy in neoadjuvant setting of locally advanced NSCLC. However, CMA markers don’t predict response to platinum-based chemo nor major pathological response (MPR). Furthermore, higher LAMP2A expression prognosis biomarker seems counterintuitive because his association with both longer DFS and OS.

In the clinical point of view, this kind of study in neoadjuvant setting of  NSCLC worth only if they can predict response to chemo and/or MPR for instance to decrease micrometastases at distant sites and tumor burden preoperatively to increase resectability rates. Futhermore, because neoadjuvant chemotherapy increased the rate of perioperative morbidity, because whatever the pCR : 15-18% of patients with stage IB/II NSCLC had no surgery if they go to neodjuvant chemotherapy !!! (https://meetinglibrary.asco.org/record/195951/abstract) ,  EMSO guidelines recommends “Surgery should be offered to all patients with stage I and II NSCLC as the preferred treatment” and “Adjuvant ChT should be offered to patients with resected stage II and III NSCLC [I, A] and can be considered in patients with resected stage IB disease and a primary tumour>4 cm [II, B].” P. E. Postmus et al., Annals of Oncology 28, iv1-iv21 (2017).

I am wondering about the fact that 41 patients with stage I and II NSCLC (40.6% of the study cohort) receive chemo in neoadjuvant setting instead of adjuvant setting. was it in context of clinical trial?

Whatever, the authors should address my 2 mains concerns:

1-Not clear if the IHC staining was performed on the tissue before the neoadjuvant treatment (diagnosis biopsy) or on the surgical specimen (after the neoadjuvant) treatment or both?

Because the clinically relevant question is: Can we predict in advanced (using CMA makers) which patient will not benefit from platinum-based chemo in neoadjuvant setting to avoid toxicity related to chemotherapy? Staining should be performed at least on tissue sample at diagnosis. If the authors use surgical specimen after neoadjuvant, they should show that platinum chemo doesn’t affect CMA markers which is quite counterintuitive. To do so, because they need to show that LAMP2A and HSPA8 IHC is correlated in paired sample of each patient (before and after the neoadjuvant treatment).

2-IHC as a sole method is not sufficient to publish in this journal, complementary analysis such as Western blot (WB) of frozen samples of surgical sample need to be perform for date validation. WB will be allowed only if the concern 1 is address (correlation between sample at diagnosis and sample after the neoadjuvant treatment). In the same manner, LAMP2A and HSPA8 at mRNA level and copy number alteration (CNA) should also be evaluated and compare with data from cbioportal to interpret (explain) your results which are in divergent with most of previous data related to CMA markers (LAMP2A).

Does molecular alterations (TP53, EGFR, KRAS mutations) asses by NGS in your samples affect LAMP2A and HSPA8 expression? Add line with molecular alteration in table 1.

Other comments

Line 48 “Yet the proportion of these types of cancer still remains low”:  I don’t agree:  approximatively 70 % of metastatic LUAD and 45% of early  LUAD harbours targetable molecular alterations ( Skoulidis et  al. Nature 2019).

Line 48 -50: “Patients also profit from immunotherapy and immune checkpoint inhibitors, which have recently been approved for unresectable stage III cancer therapy”.  In this indication the only immunotherapy approved is the ICI durvalumab which is in fact approved in consolidation after concurrent chemo-radiation (Pacific  trial).

Line 56-57: “it is generally accepted that early-stage NSCLC patients should not be treated with surgery alone”:  I don’t agree: stage I disease with tumor <4 don’t benefit from adjuvant chemo. “Surgery should be offered to all patients with stage I and II NSCLC as the preferred treatment” and “Adjuvant ChT should be offered to patients with resected stage II and III NSCLC [I, A] and can be considered in patients with resected stage IB disease and a primary tumour>4 cm [II, B].” P. E. Postmus et al., Annals of Oncology 28, iv1-iv21 (2017).

Table 1: 10% or less residual viable tumour after neoadjuvant chemotherapy, is termed major pathological response so <1% and 1-<10%  should be merged.

The status of radical surgery (R0, R1 and R2) should also be mentioned in the table.

Round 2

Reviewer 1 Report

The authors have satisfactorily responded to all my questions and made the necessary changes to the manuscript. I have no further comments.

Author Response

We thank the reviewer for the favorable evaluation of our work.

Reviewer 2 Report

I have no further comments.

Author Response

(The authors gave the same response as above.)

Reviewer 3 Report

The new Aim of the work is not clear for me and is not relevant for any aspect of lung cancer mangement (diagnosis, treatment, predictive nor  prognosis biomarker, physiopathology) including the scope of outcome improvements with neodajuvant treatment of NSLC described in introduction.

“The aim of our study was to evaluate the expression of the
CMA-associated proteins LAMP2A and its co-player HSPA8 in locally
advanced resected human NSCLC and the dependence of expression from
antecedent neoadjuvant therapy.”

I am not sure that IHC staining alone (including flaws in the experimental design) is sufficient to be publish in this journal.  Futhermore, staining samples after neodjuvant treatment (platinum based chemo) with a missing control (sample before treatment) in the context of CMA markers is an important flaws. We can not assume without any control that chemo doesn't impact CMA markers.

What about CMA makers  in non tumoral lung sample???

Author Response

We thank the reviewer for the detailed and precise evaluation of our work and the productive suggestions that had helped us so much to improve our manuscript.

Below we provide the point by point responses to the second round comments:

The new Aim of the work is not clear for me and is not relevant for any aspect of lung cancer management (diagnosis, treatment, predictive nor  prognosis biomarker, physiopathology) including the scope of outcome improvements with neodajuvant treatment of NSLC described in introduction.

“The aim of our study was to evaluate the expression of the CMA-associated proteins LAMP2A and its co-player HSPA8 in locally advanced resected human NSCLC and its dependence of expression from antecedent neoadjuvant therapy.”

Reply: We regret not being able to rewrite the Aim section to be more understandable. But we still consider the presented study relevant for potential future lung cancer management. The study design is focused on a basic data collection of the expression of novel CMA-associated biomarkers in NSCLC. There is no intention nor ambition to implement these data clinically in the very near future, but rather we provide additional information on the topic of a lesser-studied type of autophagy, CMA, and its possible evaluation in FFPE tissue. The aim has been clarified in the revised manuscript.

I am not sure that IHC staining alone (including flaws in the experimental design) is sufficient to be publish in this journal.  Futhermore, staining samples after neodjuvant treatment (platinum based chemo) with a missing control (sample before treatment) in the context of CMA markers is an important flaws. We can not assume without any control that chemo doesn't impact CMA markers.

Reply: We thank the reviewer for the important observation. Our study is based on previously extensively validated antibodies, using different engineered cell lines and immunohistochemistry and western blot analyses.

The lack of adequate diagnostic samples pre-chemotherapy for all patients are inherent in our real-world study design. We searched in the database and found that a previous sample (biopsy or cytology before the treatment) was available for only 28/52 LUAD and 28/46 LUSC. These samples were partly obtained from the metastatic tissue in the lymph node and the correlation of the marker expression in the primary versus metastasis would need to be checked in the first place and excluded as a confounding factor. Additionally, the small size of the diagnostic biopsies and cytology specimens could results in further bias due to sampling error. In order to partly overcome this limitations,  we have assembled a control cohort that was matched by tumor stage at the time of diagnosis with our control group. We have discussed the limitation mentioned by the reviewer in the Discussion of our revised manuscript and we will take this input in account in our future studies. 

What about CMA makers in non tumoral lung sample???

Reply: We thank the reviewer for this very interesting comment. In the present study, we have focused on the expression of CMA-markers in the tumor cells and we have not evaluated the basic marker expression in normal lung tissue nor in tumor associated stroma. The lung tissue around tumors is oftentimes fibrotic or heavily inflamed or shows at least some reactive changes with different adaptations of the immune cell composition or macrophage content. All those changes could possibly impact CMA expression in “non tumoral lung samples”, and the heterogeneity of different morphological aspects would add significant complexity in the evaluation of expression. Thus, we thank the reviewer for bringing up the topic of CMA marker expression in “non tumoral lung”, but deem it to be the subject of a separate study we would like to conduct.